# TDP-43 mutations link Amyotrophic Lateral Sclerosis with R-loop homeostasis and R loop-mediated DNA damage

Marta Giannini [1,2,3ʘ], Aleix Bayona-Feliu [3,4ʘ], Daisy Sproviero [1ʘ], Sonia I. Barroso [3], Cristina Cereda [1]*, Andrés Aguilera [3,4]*

1 Genomic and Post-Genomic Center, IRCCS Mondino Foundation, Pavia, Italy, 2 Department of Brain and Behavioral Sciences, University of Pavia, Pavia, Italy, 3 Andalusian Center of Molecular Biology and Regenerative Medicine-CABIMER, Universidad de Sevilla-CSIC-Universidad Pablo de Olavide, Seville, Spain, 4 Departamento de Genética, Facultad de Biología, Universidad de Sevilla, Seville, Spain

ʘ These authors contributed equally to this work.
* cristina.cereda@mondino.it (CC); aguilo@us.es (AA)

**Data Availability Statement:** All relevant data are within the manuscript and its Supporting information files and the Spreadsheet of source data.

## Abstract

TDP-43 is a DNA and RNA binding protein involved in RNA processing and with structural resemblance to heterogeneous ribonucleoproteins (hnRNPs), whose depletion sensitizes neurons to double strand DNA breaks (DSBs). Amyotrophic Lateral Sclerosis (ALS) is a neurodegenerative disorder, in which 97% of patients are familial and sporadic cases associated with TDP-43 proteinopathies and conditions clearing TDP-43 from the nucleus, but we know little about the molecular basis of the disease. After showing with the non-neuronal model of HeLa cells that TDP-43 depletion increases R loops and associated genome instability, we prove that mislocalization of mutated TDP-43 (A382T) in transfected neuronal SH-SY5Y and lymphoblastoid cell lines (LCLs) from an ALS patient cause R-loop accumulation, R loop-dependent increased DSBs and Fanconi Anemia repair centers. These results uncover a new role of TDP-43 in the control of co-transcriptional R loops and the maintenance of genome integrity by preventing harmful R-loop accumulation. Our findings thus link TDP-43 pathology to increased R loops and R loop-mediated DNA damage opening the possibility that R-loop modulation in TDP-43-defective cells might help develop ALS therapies.

## Authors' summary

Amyotrophic Lateral Sclerosis (ALS) is an adult onset, progressive neurodegenerative disease, caused by the selective loss of upper and lower motor neurons in the cerebral cortex, brainstem and spinal cord. The nuclear TDP-43 RNA binding protein, is encoded by a major gene for ALS susceptibility whose mutations are found in 3% of familial and 2% of sporadic ALS cases. Thanks to its ability to recognize DNA and RNA, TDP-43 is involved in different steps of mRNA metabolism and in several mechanisms of genome integrity. This, together with the fact that R loops or DNA-RNA hybrids are a common source of genome instability, prompted us to investigate whether TDP-43 deficiency has any role in

**Funding:** Research in C.C's lab was funded by Fondazione Regionale per la Ricerca Biomedica for TRANS–ALS (Translating Molecular Mechanisms into ALS risk and patient's well-being: FRRB 2015-0023) and in A.A.'s lab was funded by grants from the European Research Council (ERC2014-AdG669898 TARLOOP), the Spanish Ministry of Economy and Competitiveness (BFU2016-75058-P), and the European Union (FEDER). A.B.F. was supported by a Juan de la Cierva-Formación fellowship (FJCI-2017-34536) from Spanish Ministry of Science and Innovation. M.G. was recipient of a scholarship provided by Erasmus+ Mobility for Traineeships from University of Pavia. The funders had no role in study design, data collection and analysis, decision to publish, or preparation of the manuscript.

**Competing interests:** The authors have declared that no competing interests exist.

R loop homeostasis that could explain previously described DNA damage response defects of ALS cells. We show that TDP-43 plays a role in preventing R loop-accumulation and associated genome instability in neuronal and non-neuronal cells, as well as in patient cell lines. Thus, our study opens the possibility that R loop-modulation in TDP-43-defective cells might help develop ALS therapies.

## Introduction

TDP-43 is a nuclear RNA binding protein (RBP) with a repressor role of HIV-1 transcription. It binds to the trans-active response element DNA sequence of the viral genome [1, 2]. Like other hnRNP proteins, TDP-43 binds to nascent pre-mRNA molecules when they are released from the RNA Polymerase II (RNApol II) and regulates RNA maturation either through sequential interactions with or in collaboration/antagonism with specific RNA binding factors [3]. TDP-43 is also involved in the regulation of non-coding RNAs like miRNAs and lncRNAs [4, 5]. Thanks to its ability to recognize single-stranded DNA (ssDNA) or single-stranded RNA (ssRNA) with a preferential binding to (UG)n-enriched sequences [6], TDP-43 is involved in different steps of mRNA metabolism and in several mechanisms of genome integrity [7], consistent with the idea that RNA metabolism and DNA damage response (DDR) may be functionally interconnected [8].

Mutations in TDP-43 are associated with sporadic and familial cases of Amyotrophic Lateral Sclerosis (ALS), an adult onset, progressive neurodegenerative disease, caused by the selective loss of upper and lower motor neurons in the cerebral cortex, brainstem and spinal cord [9, 10]. TARDBP is a major pathological gene for the ALS susceptibility and their mutations are found in 3% of familial and 2% of sporadic ALS cases [11, 12]. Particularly, homozygous p.A382T TARDBP variation (A382T TDP-43) is one of the most common missense mutation in familial patients. A382T TDP-43 accumulation in the cytoplasm can reduce its physiological nuclear function, such as transcription regulation, mRNA splicing and transport [13–15] and miRNAs biogenesis [5, 9]. Subsequent to this, the formation of oligomers and aggregates of TDP-43 in the cytoplasm may recruit native TDP-43 or other interactors proteins [16], constituting a gain of toxic function associated with neurodegeneration [17]. TDP-43 aggregates are identified as a major component of the ubiquitinated neuronal cytoplasmic inclusions deposited in spinal motor neurons both in familiar and sporadic ALS patients [18].

In addition to transcriptional autoregulation, TDP-43 can be cleaved into smaller C-terminal fragments before being enzymatically degraded to maintain its physiological levels [9, 19] by a range of cysteine proteases, including caspases and calpains. Moreover, lines of evidence suggest that these CTFs can be produced via translation of an alternative transcript which is upregulated in ALS [20]. Recent studies proved that increased cytosolic sequestration of the poly-ubiquitinated and aggregated forms of mutant TDP-43 correlates with higher levels of DNA strand breaks, activation of DDR factors such as phospho-ataxia-telangiectasia mutated (ATM), phospho-53BP1, γH2AX in SH-SY5Y lines expressing wild-type (WT) or Q331K-mutant TDP-43 [21]. TDP-43 depletion leads to increased sensitivity to various forms of DNA damage and mutation in the C-terminus glycine-rich low-complexity region (LC domain) associates with the loss of its nuclear function [22]. In addition, TDP-43 colocalizes with active RNA polymerase II at sites of DNA damage along with the DDR protein, BRCA1, participating in the prevention and/or repair of R loop-associated DNA damage [23].

Evidence indicate that a major source of spontaneous DNA damage comes from the accumulation of R-loops, consisting in DNA-RNA hybrids and a displaced single strand DNA

(ssDNA) [8]. Non-physiological R loops occur as unscheduled events formed co-transcription-ally that can compromise genome integrity. Increasing evidence [16, 24] has highlighted a common association of increased R-loops with a variety of genetic diseases, including neuro-degenerative disorders [25]. R-loop formation is enhanced in genomic regions containing highly repetitive DNA, which could facilitate the thermodynamic stabilization of RNA-DNA hybrids [26, 27] and in cells mutated in genes encoding factors controlling R-loop homeosta-sis. Such factors are generally related to RNA processing and export or have DNA-RNA unwinding (helicase) or hybrid-specific ribonuclease (RNase H) activities [28, 29]. However, a crucial role in prevention of R-loop formation is also played by the DDR. It is particularly notorious the role of BRCA2 and BRCA1 DSB repair factors or the Fanconi Anemia pathway (FA), especially FANCD2, involved in the repair of the inter-strand crosslinks (ICLs) and rep-lication fork blockages [30, 31]. Deficiency on any of these factors lead to harmful R-loop accu-mulation in human cells [8].

All this, together with the fact that a number of neurodegenerative diseases highlight a par-ticular sensitivity of the nervous system and motor neurons are associated with deficiencies in RNA metabolism and DDR, prompted us to investigate whether TDP-43 deficiency, as found in ALS cells, have a role in R-loop homeostasis that could explain previously described DDR defects of ALS cells. We show that TDP-43 plays a role in preventing R-loop accumulation and R loop-mediated DNA breaks in neuronal and non-neuronal cells and in patient cell lines, thus opening the possibility that R-loop modulation in TDP-43-defective cells might help develop ALS therapies.

## Results

### TDP-43 depletion causes R loops, DNA damage and FANCD2 repair centers in HeLa cells

A key regulatory role of TDP-43 in essential metabolic processes was previously suggested since silencing of TDP-43 in HeLa cells lead in dysmorphic nuclear shape, misregulation of the cell cycle, apoptosis, increase in cyclin-dependent kinase 6 (Cdk6) transcript and protein levels [32]. As a major readout associated with RNA transcription metabolic defects, we ana-lyzed accumulation of nuclear DNA-RNA hybrids in TDP-43 depleted HeLa cells (siTDP-43 HeLa cells) as a reference cell line commonly used in R loop and genome integrity studies.

Genomic DNA-RNA hybrids in siTDP-43 HeLa cells were first assessed by immunofluores-cence microscopy (IF) using the anti-DNA-RNA hybrid S9.6 antibody, and determining the levels of the S9.6 signal in the nucleoplasm after subtracting the nucleolar contribution [33, 34]. As controls we used HeLa cells transiently transfected with a mock control vector express-ing GFP (siC) or overexpressing the RNaseH1 enzyme, which specifically degrades the RNA moiety of hybrids. A slight but significant increase of R loops was observed in siTDP-43 HeLa cells, in which TDP-43 protein levels were reduced 75% (S1A Fig), in comparison to the siC (Fig 1A). Efficient RNaseH1 overexpression from the pEGFP-M27 plasmid, as confirmed by IF (S1B Fig), reduced significantly the S9.6 signal, confirming that the signal detected corre-sponded to R-loops (Fig 1A). A comparative analysis of the S9.6 signal intensity obtained for depletion of other cellular factors that protect cells from R loops in HeLa cells shows that the signal increase after TDP-43 depletion was similar to that obtained by depletion of other fac-tors such as THOC1, UAP56, SETX, AQR, DDX23 mRNP processing factors (S1C Fig).

Next, we determined R-loop accumulation by the more accurate method of DNA-RNA immunoprecipitation (DRIP)-qPCR, based specifically on the purification of genomic DNA-RNA hybrids of different sizes. In this case the S9.6 signal was determined for genes expressed at different levels such as APOE, RPL13A, WDR90, EGR1 and MIB2, which have

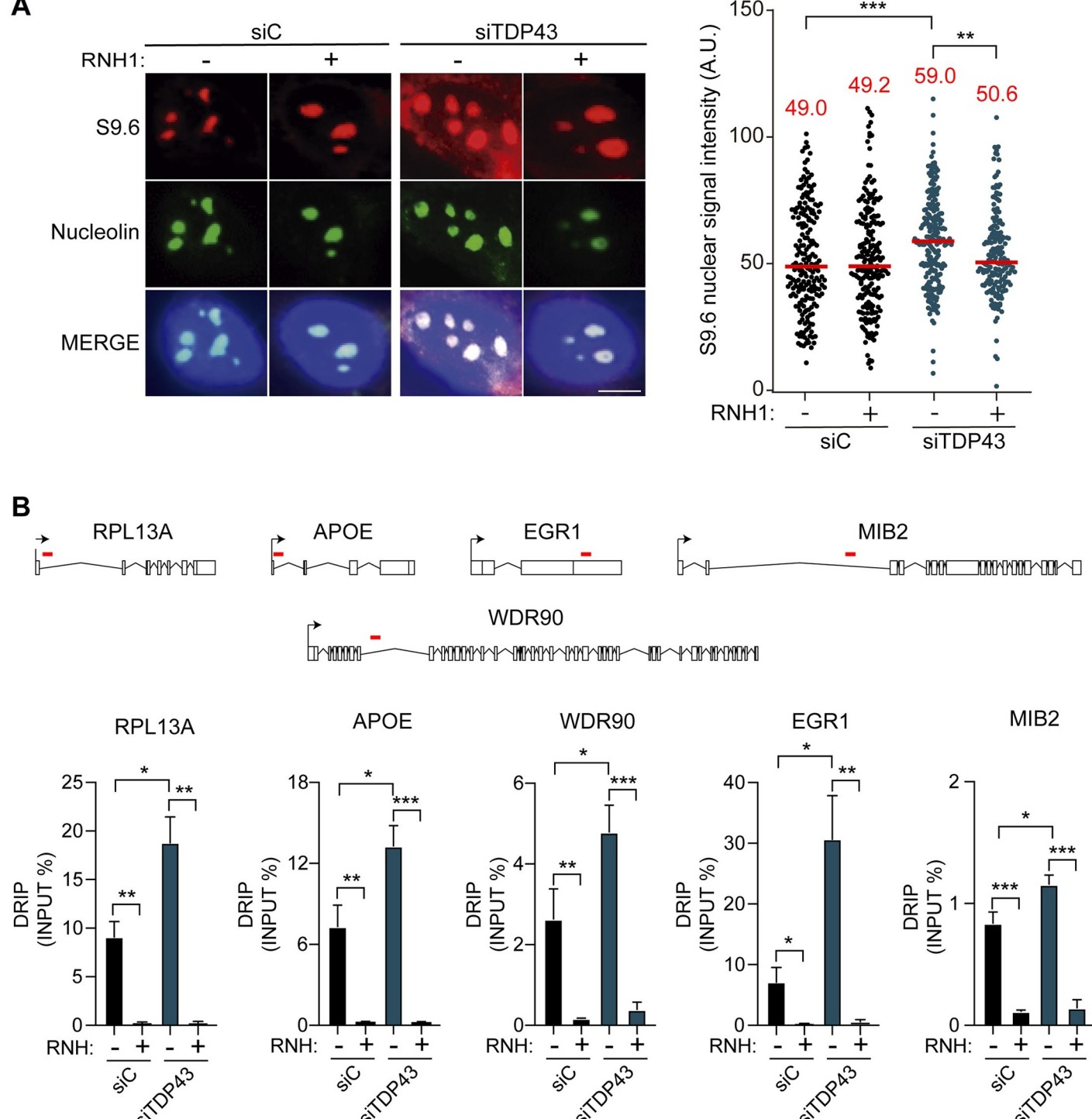

**Fig 1. R-loops accumulation in siTDP-43 HeLa. A)** siC and siTDP-43 HeLa immunostaining with antiS9.6 antibody and anti-nucleolin antibody. The graph shows the median of the S9.6 intensity per nucleus after nucleolar signal removal. Around 300 cells from three independent experiments were considered. Scale bar: 25μm. ***, P < 0,0002; **, P < 0,001 (Mann-Whitney U test, two-tailed). **B)** DRIP-qPCR using the anti S9.6 antibody at RPL13A, APOE, WDR90, EGR1 and MIB2 are shown in siC HeLa and siTDP-43 HeLa. Pre-immunoprecipitated samples were untreated (-) or treated (+) with RNaseH, as indicated. Data represent mean ± SEM from three independent experiments. *, P <0,05, **, P < 0,01, ***, P < 0,001 (Upaired t test, one-tailed). In all cases, when no asterisk is shown indicates that is not significant.

been previously validated for the detection of R loops [31, 33, 35], and the poorly expressed SNRPN gene used as negative control [35, 36]. We detected accumulation of DNA-RNA hybrids in the analysed genes in siTDP-43 HeLa cells compared to the siC HeLa cells, obtaining a significative increase on all genes tested (Fig 1B), whereas the SNRPN negative control did not show R loop accumulation (S1D Fig), further supporting the validity of our DRIP-qPCR methodology for R loop detection. Importantly, RNaseH treatment induced a dramatic signal decrease confirming that signals were R-loop specific (Fig 1B).

Then, we investigated the functional impact of nuclear DNA-RNA hybrid enrichment on DDR, given that hybrids have been shown to enhance transcription-replication conflicts [37]. As can be seen in Fig 2A, γH2AX foci, as determined by IF, were significantly increased in siTDP-43 compared to siC HeLa cells. γH2AX foci significantly decreased after RNaseH1 overexpression, indicating that the damage caused by TDP-43 depletion is R-loop mediated. It

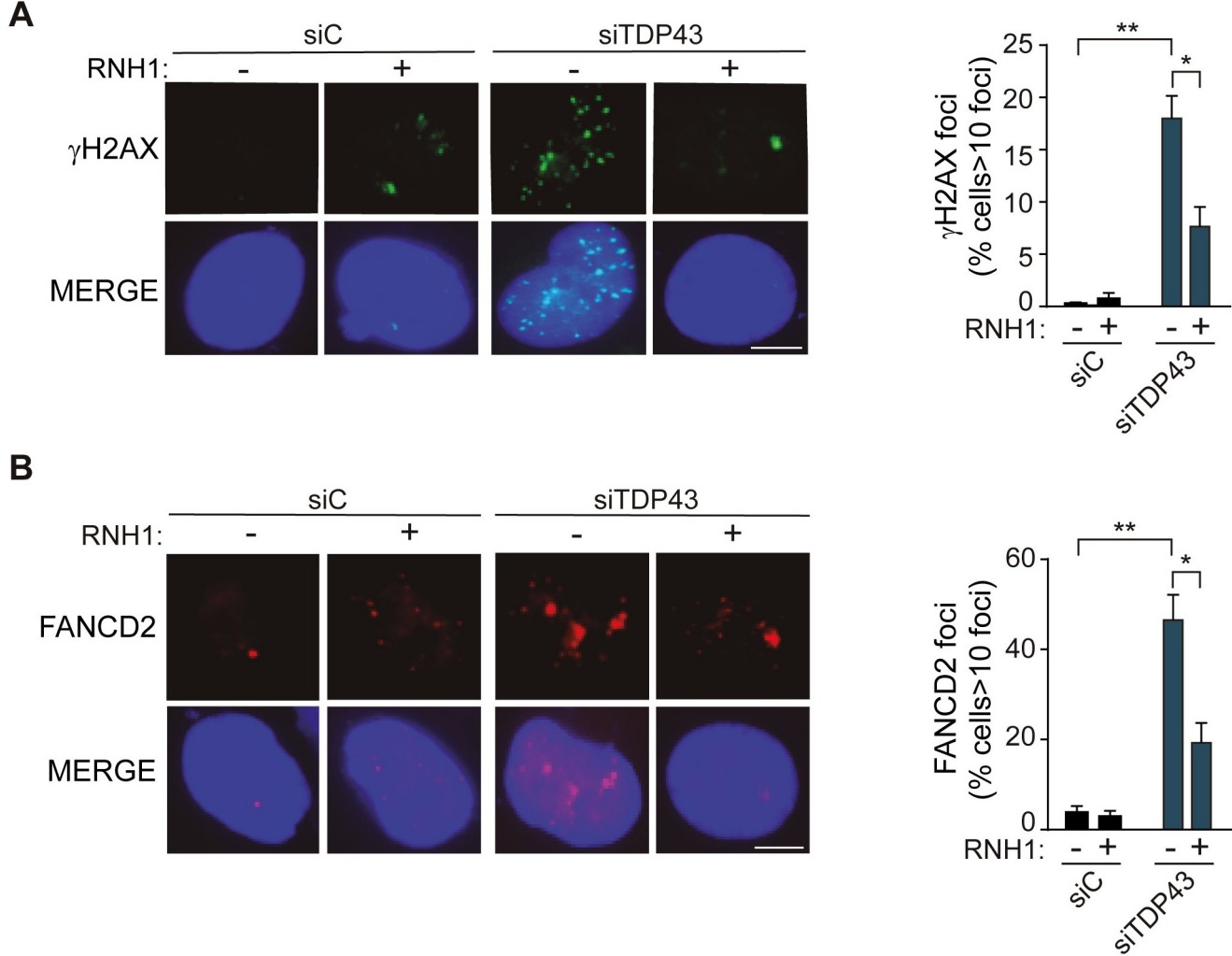

**Fig 2. TDP-43 associates with FANCD2 and affects genome integrity in siTDP-43 HeLa. A)** Detection of γH2AX foci by IF in siC and siTDP-43 HeLa. The histogram shows the quantification of the relative amount of cells in percentage containing >10 γH2AX foci in each case. More than 100 cells were counted in each of the three experiments. Data represent mean ± SEM from three independent experiments. Scale bar: 25μm. **, P<0,01 and *, P< 0,05 (Upaired t test, two-tailed). **B)** Detection of FANCD2 foci by IF in siC and siTDP-43 HeLa. The histograms show the quantification of the relative amount of cells in percentage containing >10 FANCD2 foci in each case. Details as in Fig 2A. Scale bar: 25μm. **, P<0,01 and *, P< 0,05 (Upaired t test, two-tailed). In all cases, when no asterisk is shown indicates that is not significant.

has been shown that the Fanconi Anemia (FA) repair pathway is a critical pathway to resolve R loop-mediated DNA breaks as the result of transcription-replication collisions and that the FA factors work at the collisions [30, 31, 38–40]. Therefore, we tested whether the damage generated by TDP-43 depletion was signaled by the FA pathway, for which we used the FANCD2 component [38]. As it can be seen in Fig 2B, FANCD2 foci were significantly increased in siTDP-43 HeLa cells compared to the siC control. Importantly, this increase was reduced by RNaseH1 overexpression, proving that TDP-43 depletion is responsible for an accumulation of the Fanconi Anemia repair factor caused by R-loop accumulation. The result is consistent with the idea that FANCD2 accumulates at R loop-containing sites at which the replication fork is blocked, similarly to inactivation of other RNA metabolic factors that lead to R-loop accumulation [38, 41].

## Genome-wide co-localization of TDP-43 at expressed genes

Our results show that TDP43 depletion causes R loop-accumulation, R loop-dependent DNA damage and accumulation of the transcription-replication collision-associated FANCD2 repair foci similar to depletion of other mRNP processing factors that function co-transcriptionally together with RNA polymerase II (RNAPII). Indeed, genome-wide ChIP-seq data obtained in K562 erythroblastoma cells (ENCODE Project; ENCSR033VAZ entry) reveals that TDP-43 colocalizes with expressed genes defined by RNA-seq (Fig 3A). 6011 out of 6245 of genes recruiting TDP-43 correspond to actively expressed genes (Fig 3B). Consistently, TDP-43 occupancy is significantly higher in expressed genes compared to genes with low or non-detectable expression, with a preference towards the 5' end of the genes (Fig 3C and 3D). Interestingly, analysis of the genes reported to be prone to accumulate R loops in K562 cells [42] reveals that 4809 of the 6245 genes are enriched in TDP-43. These results support that the TDP-43 RBP is present at expressed genes that are prone to accumulate R loops, where it might participate in mRNP biogenesis, similarly as proposed for other co-transcriptional RNA binding factors [8]. A remaining question is which specific role TDP-43 may play during transcription of those genes.

## Cytoplasmic mislocalisation of mutated TDP-43 causes R-loop accumulation and leads to activation of the DDR and the Fanconi Anemia pathway

In ALS patients harboring TDP-43 mutations, TDP-43 mislocalizes from the nucleus to the cytoplasm in detergent-resistant aggregated forms either full-length (43 KDa) and fragmented forms (35KDa, 25KDa), which can be ubiquitinated and hyperphosphorylated [43]. We hypothesized that TDP-43 mislocalization due to missense mutations could have an impact on R-loop accumulation and DNA damage in ALS disease. For this we moved our studies to the SH-SY5Y neuroblast-like cells usually used as in vitro models of neuronal function and differentiation and to assay ALS related mutations. To determine TDP-43 cellular localization, we performed IF microscopy in basal SH-SY5Y, SH-TDP+ (overexpressing a GFP-tagged TDP-43 WT form), SH-TDP382 (expressing the GFP-tagged p.A382T TDP-43 mutant form) and SH-TDP294 (expressing the GFP-tagged p.G294V TDP-43 mutant form) cells using an anti-TDP-43 or anti-GFP antibody, able to detect the wild-type nuclear protein and the cytoplasmic full length and fragmented forms. In all cases, TDP-43 overexpression levels, as determined by Western were similar (S2A Fig), excluding the possibility that a potential different phenotype could be attributed to different TDP-43 levels rather than the dysfunction caused by the mutation itself. Western blot analysis of nuclear and cytoplasmic fractionation showed an increase level of GFP-fused proteins in the cytoplasm of SH-TDP+ and SH-TDP382, reducing the total nuclear fraction, including endogenous TDP-43 and GFP-fused, with respect to the

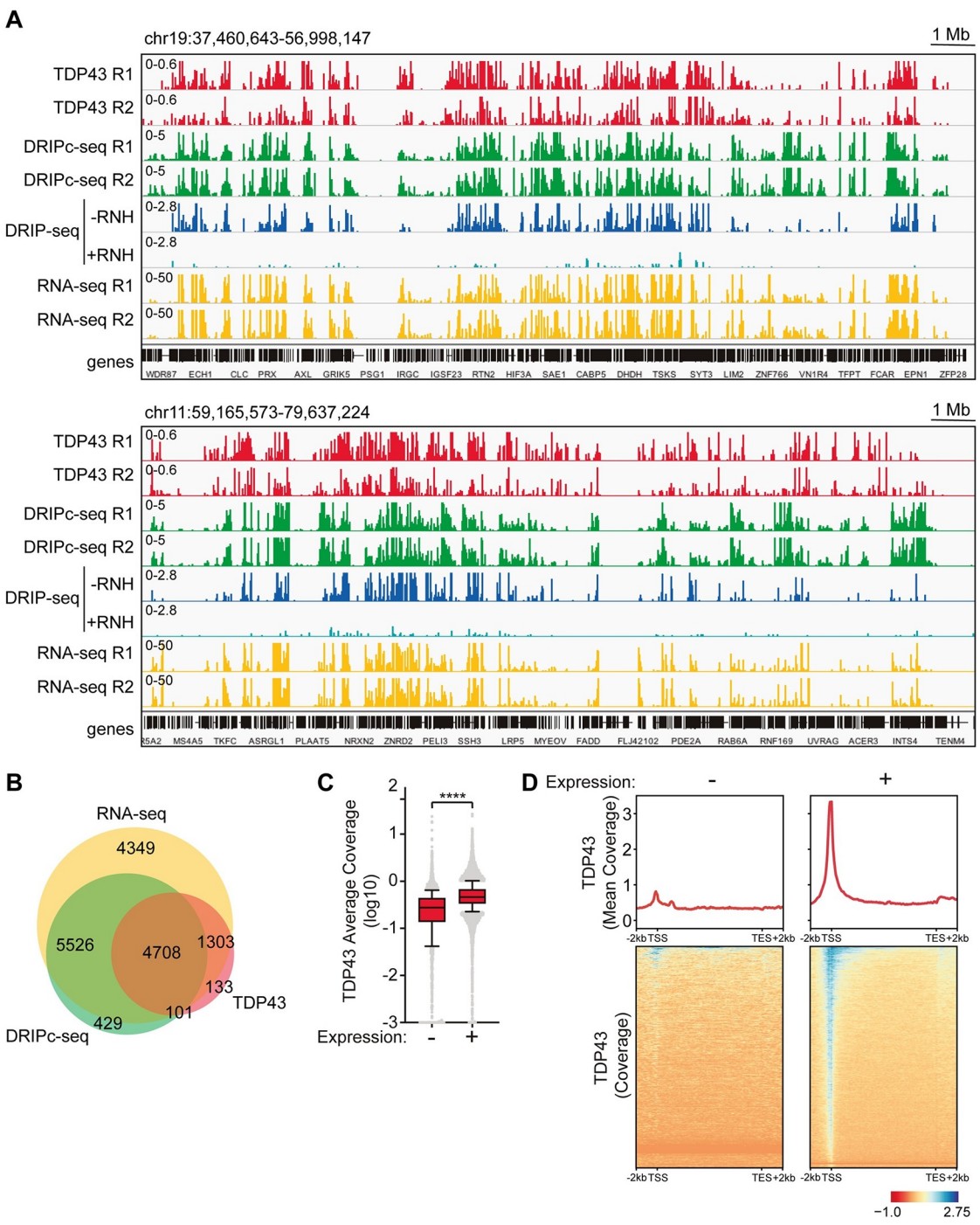

**Fig 3. TDP-43 co-localizes genome-wide with expressed R loop-prone genes. A)** Screenshots of genome windows showing co-occurrence between TDP43 (red), DRIPc-seq (green), untreated (dark blue) and RNH-treated (light blue) DRIP-seq and RNA-seq (yellow) data. Genome location is indicated in the top left corner and scale in the top right. Coverage scale indicated in the top left of each track. Biological replicates are also indicated (R1, R2). **B)** Venn diagram showing correlation between genes bind by TDP-43 (TDP43; red circle), genes forming R loops (DRIPc-seq; green) and expressed genes (RNA-seq; yellow). Numbers refer to genes co-occurring between conditions. **C)** TDP-43 ChIP-seq average coverage (log10) across silent (-) and expressed (+) genes. ****, P < 0,0001 (Mann-Whitney U test, two-tailed). **D)** Metagene analysis showing TDP-43 ChIP-seq coverage (red line) along gene body (+/- 2kb) of silent (-) and expressed (+) genes. Mean coverage is plotted in the upper panel and heatmap intensities for the entire gene population is shown below. Scale is also indicated.

cytoplasmic fraction (S2B Fig). In the case of SH-TDP294, the ratio between nuclear and non-nuclear signal did not show the same tendency, raising the possibility that protein dysfunction can be causative of the phenotype. However, in all cases, SH-TDP+, SH-TDP382 and SH-TDP294 the cytoplasmic fraction of the GFP-overexpressed protein was clearly higher compared to the SH-SY5Y control, consistent with a cytoplasmic mislocalization. From now we focused our study in the A382T mutant.

We confirmed by flow cytometry and IF that overexpression of both TDP+ and TDP382 occurred at similar levels (S2C and S2D Fig). In addition, in SH-TDP+, the RBP was localized preferentially in the perinuclear area compared to non-transfected SH-SY5Y cells, in which localization was predominantly nucleoplasmic. TDP-43 nuclear localization was significantly decreased in SH-TDP382 cells compared both to non-transfected SH-SY5Y and SH-TDP+ (S2E Fig). These changes could not be attributed to differences in nuclear area as these did not show any significant difference (S2F Fig), confirming that the A382T mutation is linked to TDP-43 cytoplasmic mislocalization with formation of inclusions or aggregates as previously reported for this and other mutations [44–46]. Results were the same when using the anti-TDP43 or anti-GFP antibodies.

Next, we tested whether mislocatization of TDP-43 could impact onto genomic integrity and R-loop accumulation in these cells. We first assayed whether overexpression of wild-type TDP-43 and mutated and mislocalized TDP-43 affected R-loop accumulation. Since both mutated and overexpressed TDP-43 may affect its physiological role in miRNA biogenesis, we added in this case an additional treatment with RNaseIII, which degrades specifically dsRNAs, to counteract the reported ability of S9.6 to detect dsRNAs [47, 48]. A significant increase of nucleolar S9.6 intensity was detected both in SH-TDP+, SH-TDP382 and SH-TDP294 cells compared to SH-SY5Y (Fig 4A and S3A Fig). Furthermore, RNaseH1 overexpression, as confirmed by IF (S3B Fig) caused nuclear S9.6 signal decrease in both SH-TDP382 and SH-TDP294 cells, but not in SH-TDP+ cells. Therefore, we conclude that ALS TDP-43 mutations also cause R-loop accumulation. From now on we focused our study on the TDP-43 A382T mutant protein as a representative mutant protein linked to ALS.

We performed DRIP-qPCR in the neuroblastoma cell lines in previously validated genes for R loop detection [33, 35, 36], to confirm the results involving TDP-43 role in preventing R-loop accumulation in human cells. Consistent with IF results a statistically significant increase in R loops was detected by DRIP-qPCR in RPL13A, WDR90 and EGR1 genes in SH-TDP382 cells, even though not for APOE likely due to an unknown cell type-specific effect (Fig 4B). The levels were minimal for the SNRPN negative control (S3C Fig). RNaseH treatment dramatically decreased the levels of the signal in all cases, confirming that the signal detected was specific for nuclear DNA-RNA hybrids. Results were thus consistent with those obtained in TDP-43-depleted HeLa cells. Interestingly, an increase in the immunoprecipitated material was also detected in some genes in SH-TDP+ cells (Fig 4B). In this sense, it is worth noting that overexpression of wild-type TDP-43 has been reported to be detrimental to cells [49], and also led to a minor but significant decrease in nuclear TDP-43 abundance (S2B Fig). This could explain the minor but significant R-loop increase observed by DRIP at some genes in SH-TDP+, consistent with the high R-loop levels correlating with a low nuclear TDP-43 content, as is the case of SH-TDP382 cells.

To test whether R-loop accumulation observed in SH-TDP382 cell lines causes DNA damage we determined the levels γH2AX foci by IF microscopy and their dependence of R-loops by testing whether RNaseH1 overexpression reduced them. As can be seen in Fig 5A, γH2AX foci were significantly increased in SH-TDP382 cells whereas this was not the case in SH-TDP+ cells expressing the wild-type form of TDP-43. The increase in damage was suppressed by RNaseH1 overexpression, indicating that DNA break accumulation caused by mutant TDP43 was mediated by DNA-RNA hybrids.

**A**

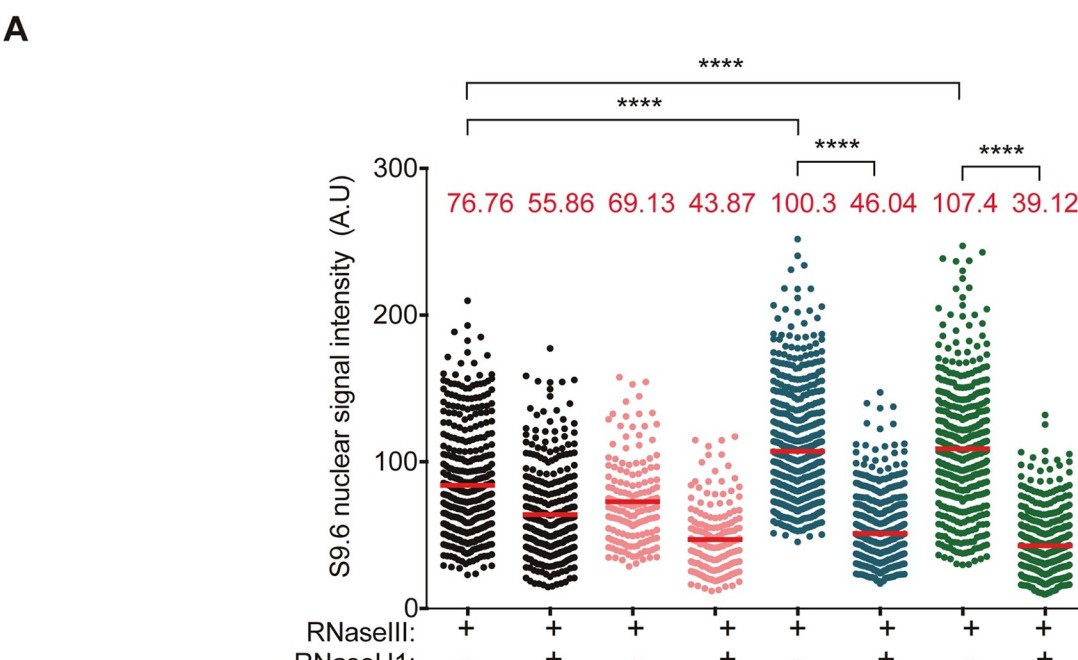

**B**

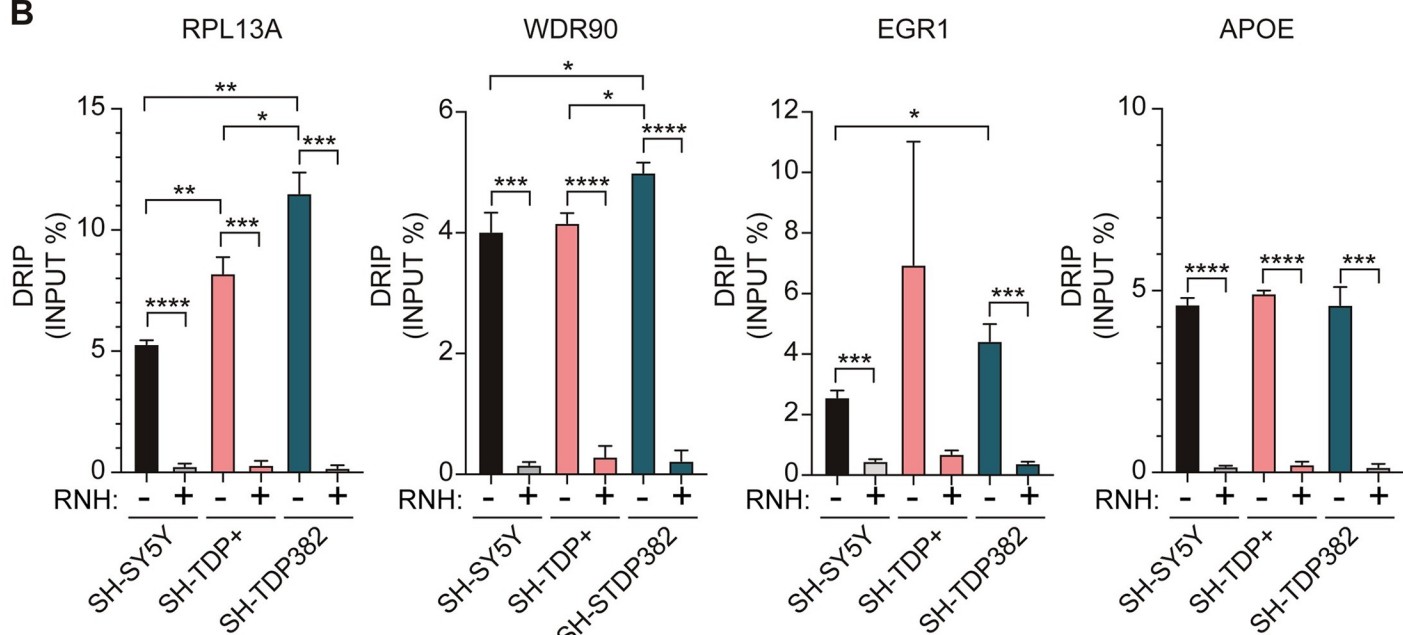

**Fig 4. R-loops accumulation in SHSY-TDP382 and SHSY-TDP294 overexpressing SH-SY5Y cells. A)** IF using anti S9.6 antibody and anti-nucleolin antibody of SH-SY5Y, SH-TDP+, SH-TDP382 and SH-TDP294. The graph shows the quantification of S9.6 nuclear signal. Details as in Fig 1A. ****, P < 0.0001 (Mann-Whitney U test, two-tailed). **B)** DRIP-qPCR using S9.6 antibody, at RPL13A,WDR90 and EGR1 genes in basal SH-SY5Y, SH-TDP+, SH-TDP382 and SH-TDP294. Details as in Fig 1B. *, P < 0,05; **, P < 0,01; *** P < 0,001; ****, P < 0,0001 (Unpaired t test, one-tailed). In all cases, when no asterisk is shown indicates that is not significant.

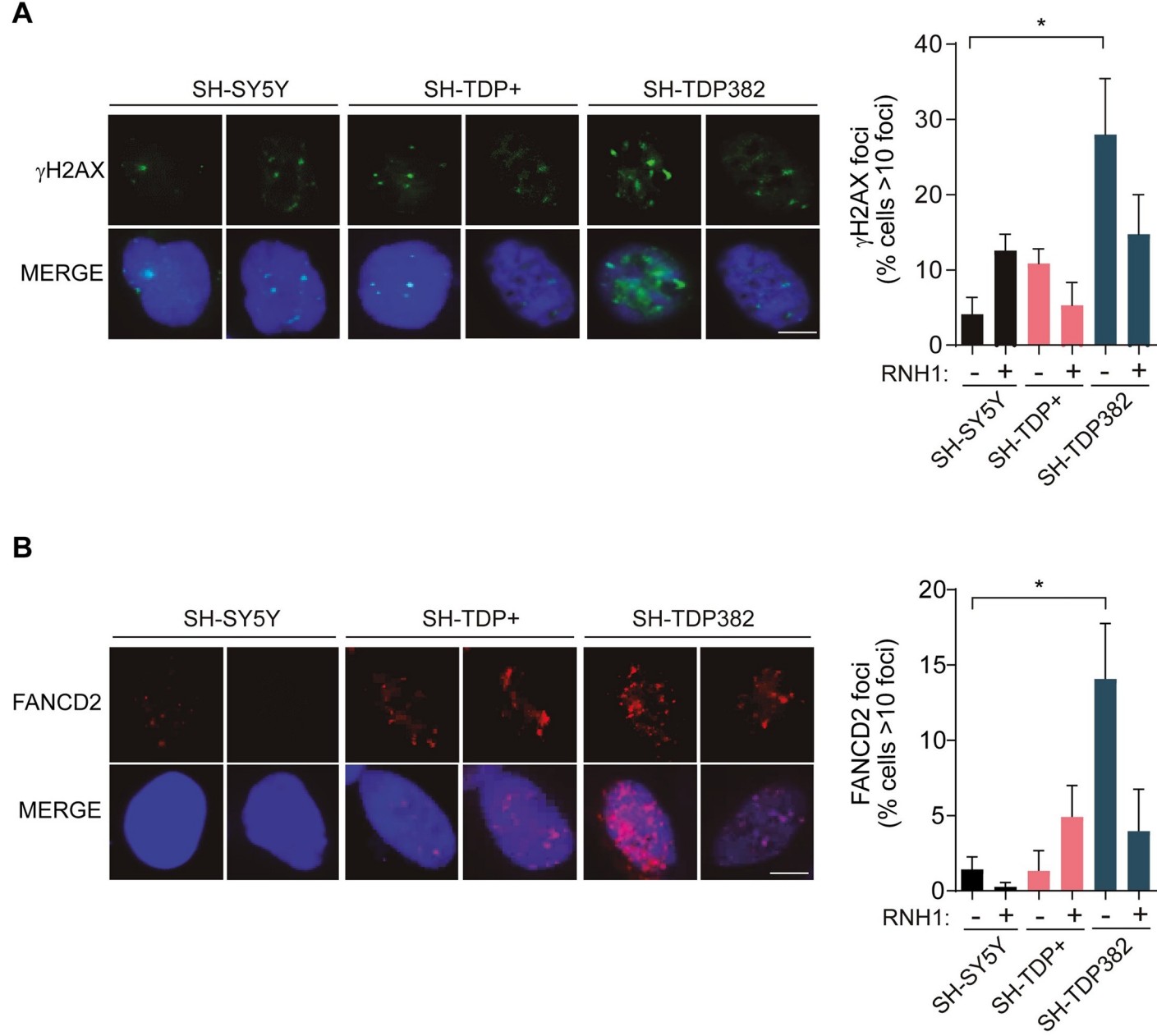

**Fig 5. TDP-43 associates with FANCD2 and affects genome integrity in stable transfected SHSY-TDP382-. A)** Detection of γH2AX foci, by IF in SH-SY5Y, SH-TDP+ and SH-TDP382. The histogram shows the quantification of the relative amount of cells in percentage containing >10 γH2AX foci in each case. Details as in Fig 2A. Scale bar: 25μm. *, P < 0,05 (Upaired t test, two-tailed). **B)** Detection of FANCD2 foci by IF in SH-SY5Y, SH-TDP+ and SH-TDP382. The graph shows the quantification of the relative amount of cells in percentage containing >10 FANCD2 foci in each case. Details as in Fig 2A. Scale bar: 25μm. *, P < 0,05 (Upaired t test, two-tailed). In all cases, when no asterisk is shown indicates that is not significant.

Next, we tested whether the origin of such DNA damage was due to an increase in transcription-replication collisions enhanced by R-loops. We determined the levels of FANCD2 foci as previously reported. Notably, FANCD2 foci were significantly increased in SHSY-TDP382 mutant cells compared to SHSY-TDP+ and this increase was reduced by RNaseH1 overexpression (Fig 5B). Therefore, the ALS pathogenic TDP-43 mutation in the analysed neuronal model leads to a comparable functional effect to that observed in silenced HeLa cells. The

pathogenic TDP-43 mutation causes an increase in DNA breaks derived from R-loop accumulation that promotes transcription-replication collisions that are processed by the FA pathway, as reported for other cases of recombinogenic R-loops [30, 31, 38–40].

## Accumulation of R-loops in p.A382T TDP-43 mutated lymphoblastoid cell lines

Next, we used lymphoblastoid cell lines (LCLs) that derive from a TDP-43 mutated patient carrying p.A382T mutation (LCL-TDP382), a sporadic ALS patient (LCL-SALS) and a healthy control (LCL-CTL) (see Materials and methods) to confirm the role of TDP-43 in R-loops removal in ALS. We performed IF microscopy for S9.6 and TDP-43 in the three cell lines mentioned with two fixation methods, methanol (Fig 6A) and paraformaldehyde (S4A Fig). R-loop quantification of the IFs confirmed a significant R-loop accumulation in LCL-TDP382 cells using both fixation methods. Concomitantly, both fixation methods guaranteed a decreased detection level of TDP-43 signal in the nucleus of LCL-TDP382 in comparison to LCL-CTL and LCL-SALS was also detected (Fig 6A and S4A Fig). This is in accordance with the mislocalisation of the mutated protein in the cellular cytoplasm, causing the loss of its nuclear physiological function [50]. Again, all these changes in nuclear content could not be attributed to changes in nuclear area as this remains the same (S4B and S4C Fig). Moreover, there is a colocalization of S9.6 signal with TDP-43 in the perinuclear area of LCL-TDP382 cells in comparison to LCL-CTL and LCL-SALS. R-loop quantification was also determined in LCLs by flow cytometry, in which case the analysis reported an increased positivity of S9.6 intensity in the orange peak associated with LCL-TDP382, in comparison to the blue peak associated with LCL-CTL (Fig 6B). The positive signal in LCL-TDP382 represented by the orange peak was clearly suppressed by RNase H treatment in the same sample detected as green peak, confirming that the detected signal corresponds to DNA-RNA hybrids (Fig 6B).

Finally, we investigated the possibility that TDP-43 could have a role on DNA-RNA hybrid-enriched chromatin, in which case we should expect some kind of physical association. Therefore, we wondered whether TDP-43 and genomic DNA-RNA hybrids colocalize by performing a co-immunoprecipitation (coIP) in chromatin (Chr) fractions from the three cell lines (S5A Fig). At the same time, we extracted whole lysate (WL) fractions from the same samples as control (S5B Fig). In the Chr fraction, co-immunoprecipitation could be observed with the S9.6 antibody. In LCL-TDP382, the TDP-43 mutant protein showed lower levels of co-IP, while in the WL fraction of the same sample the co-IP signal was higher than the CTRL-LCLs (S5B Fig), which suggests that the mutant full length TDP-43 was not able to interact with R loop-enriched chromatin due to its sequestration at the cytosolic compartment in the cell [50], as seen before (Fig 6A and S4A Fig). Interestingly, the truncated TDP-35 form detected in Chr fraction of LCLs show high levels of S9.6 co-IP in the WL fraction of LCL-TDP382 in comparison with control LCL-CTL and LCL-SALS. This specific C-terminal mutation may predispose TDP-43 to fragmentation into CTFs, which as reported in literature are transported out of the nucleus and accumulated into complexes with RNA transcripts [49].

## Discussion

Mislocalization of TDP-43 causes a gain of neurotoxic function characteristic of the neurodegeneration process in ALS patients [51]. Moreover, TDP-43 aggregation induced by double-site mutations and TDP-43 knockdown have a common set of differentially expressed proteins, behaving in a similar way [52]. Here we show that mislocalization of mutated TDP-43 can sequester full length TDP-43 form in cytoplasmic inclusions preventing its physiological nuclear function. Importantly, this function is related to R-loop homeostasis. Nuclear

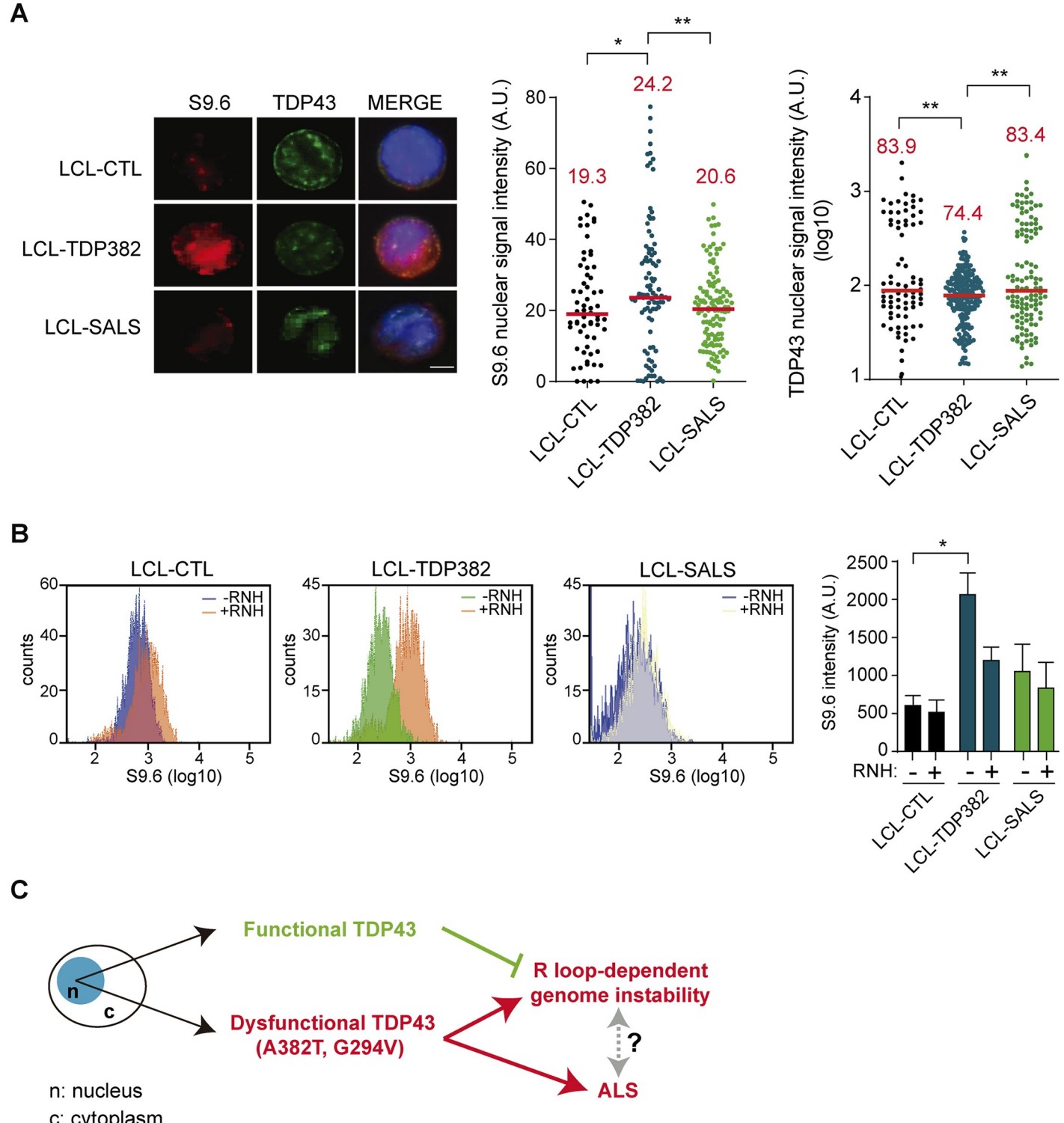

**Fig 6. R-loops accumulate in p.A382T TDP-43 mutated LCLs. A)** IF of LCL-CTL, LCL-TDP382, LCL-SALS using an anti-TDP-43 antibody and an anti-S9.6 antibody after methanol fixation. The scatter plots show the increase of S9.6 signal intensity and the decrease in TDP-43 nuclear content in SH-TDP382. Median values are indicated. Scale bar: 25μm. *, P< 0,05 (Mann-Whitney U test, two-tailed). When no asterisk is shown indicates that is not significant. **B)** Flow cytometry plot reports the amount of R-loops in LCLs (blue) in comparison to LCL-TDP382 (orange) and LCL-SALS (blue). The RNaseH action on LCL-TDP382 decreases the presence of R-loops signal (orange peak vs green peak). Histogram shows S9.6 mean fluorescence of LCL-CTL, LCL-TDP382, LCL-SALS in presence (+) and in absence (-) of RNaseH. The value represented is the mean ± SEM of three biological experiments. ANOVA, Newman-Keuls Multiple Comparison Test, *P <0,05. When no asterisk is shown indicates that is not significant. **C)** Model showing the link between TDP-43 and R loop metabolism. Functional TDP43 activity in the nucleus protects genome integrity by preventing R-loop accumulation. However, TDP-43 nuclear dysfunctions results in R loop-dependent DNA damage that eventually might lead to genome instability that could aggravate ALS phenotype.

depletion of TDP-43, as achieved by either mislocalization to the cytoplasm or siRNA depletion in different cell types, causes a significant increase in harmful R-loops that leads to DNA breaks and FANCD2 foci. The results suggest that the TDP-43 RNA-binding protein has a key role in preventing R-loop accumulation as a safeguard of genome integrity.

Silencing TDP-43 by siRNA in HeLa cells led to a significative increase of R-loop signal by S9.6 IF compared to the siC control. This was confirmed by reversion of signal in case of RNaseH1 overexpression. DRIP-qPCR revealed an important R-loop presence on 5 protein-encoding genes tested. These include RPL13A ribosomal protein gene, whose deficiency could lead to alteration of protein homeostasis and RNA metabolism [53]. In this sense, it is worth noting that TDP-43 interacting protein networks have been shown to include RPL13A [54] and that both C9orf72 mut ALS patients' derived iPSCs and TDP-43-EGFP overexpressing iPSCs presented a set of commonly destabilized RNAs involved in the ribosomal pathway [55]. However, there is no evidence that these features have any relation to R loop accumulation at RPL13A; indeed, it is a phenomenon observed in non-ribosomal protein genes also (Fig 1).

The absence of nuclear TDP-43 affects the DDR, consistent with previous reports [23]. We showed that siTDP-43 HeLa cells had a significative increase of DSBs as determined by γH2AX foci. Importantly, this DSB increase was R loop-dependent, as it could be fully reverted by RNaseH1 overexpression, and it was accompanied by an accumulation of FANCD2 foci that was also R loop-dependent. The result indicates that TDP-43 prevents the co-transcriptional accumulation of harmful R-loops that promote transcription-replication conflicts that have to be resolved by the FA pathway, consistent with the previously reported role for the FA pathway [37]. These results are particularly meaningful in the context of the putative role of TDP-43 in nuclear RNA processing, yet to be properly defined. It has been well-established that depletion of other factors involved in nuclear RNA metabolism such as the THO complex, UAP56, AQR, DDX23 or SRSF1 causes similar phenotypes as those described here for TDP43 mutations [8]. They all cause R loop accumulation and R loop-dependent genome instability that in cases like THO- and UAP56-depleted cells have been shown to correlate with an increase in FANCD2 foci and transcription-replication conflicts [38, 42]. Indeed, a very recent report that came out while ours was under review shows by DNA combing that siTDP43 cells undergo replication stress [56], consistent with our view. As it happens with those other RBP factors, genome-wide analysis of TDP-43 occupancy correlates with transcriptionally active genes, in which most of R loops accumulate (Fig 3). As shown by other mRNA processing factors, a paradigm of which is the THO complex, we propose that it is the suboptimal action of TDP-43 as an RBP during transcription what is linked to the appearance of R loops (Fig 6C). However, further research is required to completely understand how RBPs and a number of DDX proteins protect from R loops.

Notably, our assay of the impact of the TDP-43 pathogenic ALS mutation and overexpression in a neuroblastoma cell line, SH-SY5Y, revealed that the pathogenic A382T and G294V mutations in SH-SY5Y cells also affected the TDP-43 role controlling R-loop homeostasis, leading to a higher detection of genomic RNA-DNA hybrids, as detected by IF and DRIP-qPCR. As expected, DSBs and replication blockage detected by γH2AX and FANCD2 foci, respectively, were also increased in an R loop-dependent manner.

The effect of TDP-43 deficiency in R-loop homeostasis could be related to the accumulation of aberrant transcripts and hybrids that are trapped in persistent RNA-processing foci present in the cytoplasmic compartment of cell previously reported [57]. Our analysis of ALS patient-derived LCLs as a valid cellular model to study the disease that carries typical features of degenerating MNs in ALS (i.e. protein aggregation, mitochondrial disfunction etc.) [46], also revealed an increased level of nuclear RNA-DNA hybrids in LCL-TDP382 as well as co-localization of S9.6 antibody with a fraction of TDP-43 in the perinuclear area. It is possible that either sequestering of the misfolded and mutated form of TDP-43 in inclusions could cause a loss of its

nuclear function or the formation of TDP-43 aggregates in the cytoplasm could recruit native TDP-43 or other interactor proteins constituting a gain of toxic function [58]. The detection of higher S9.6 signal in both in LCL-SALS and of LCL-TDP382 underlines that R-loops may be a general condition in ALS, potentially improving RNA metabolism dysregulation and neurotoxicity that appear to be major contributors to the pathogenesis of this neurodegenerative disease.

It is worth noticing that RNAse III treatment was required to detect RNH-sensitive S9.6 reactivities (Fig 4A). Knowing that S9.6 can also detect dsRNAs [47, 48] this result indicates that TDP-43 overexpression also leads to an accumulation of dsRNA molecules. Interestingly, TDP-43 has been reported to co-localize with Dicer and Ago2, but their interaction is inhibited by aggregates formation in response to cellular stressors or also by overexpression of human TDP-43 [58]. There is no evidence that overexpression and mutation of TDP-43 lead to double hairpin pre-miRNA accumulation in neuronal models, so it is formally possible that dsRNA accumulation observed in SH-SY5Y overexpressing TDP-43 could be associated to inhibition of Dicer processing function responsible for the loss of maturation of pre-miRNAs, presented as dsRNA hairpin structures in mature miRNAs. However, it may also be possible that excess of RBPs would prevent normal RNA metabolism by excess of cellular RBPs that would bind to any RNA molecule having secondary structure segments. A negative impact of RBP overexpression on RNA metabolism has been reported in other cases [59]. These are possibilities to explore in the future.

The chromatin fraction of TDP382 mutated LCLs showed a weaker association between TDP-43 and the S9.6 signal not observed in whole cell extracts, consistent with a lower TDP-43 presence in the nucleus. This decrease was not either observed in cells from a sporadic ALS patient (Fig 6). It is known that in ALS pathological conditions TDP-43 can generate CTFs such as 35 kDa fragments both upon cleavage by caspases at intrinsic caspase cleavage sites [11], both by via translation of upregulated alternative transcript [20]. Due to the lack of nuclear localization signal (NLS), TDP-35 CTFs mislocalize to the cytoplasm, where may associate with RNA forming cytoplasmic inclusions [60]. Indeed, the biochemical analysis suggests that TDP-35 facilitates aggregate assembly promoting inclusion formation [61] and might transport different types of RNA structures. TDP-35 can also recruit full-length TDP-43 to cytoplasmic deposition from functionally nuclear localization [62] and TDP-43 continuously shuttles between nucleus and cytoplasm in a transcription-dependent manner [63]. The higher TDP-35-S9.6 co-IP in the WL fraction of LCL-TDP382 compared to the other LCLs likely may reflect inclusions formed by dsRNAs.

It is becoming clear that impaired RNA regulation and processing is a central feature in ALS pathogenesis. Our study, reinforces the need of understanding the specific role of ALS in RNA metabolism, and in particular in cells defective in the TDP-43 RBP, beyond its effect on the formation of RNA inclusion bodies in the cytoplasm. Even though this is a common readout of ALS, our study showing an increase in genomic R-loops and DNA damage and Fanconi Anemia foci, expected for obstacles blocking replication, suggests that an important cause of the disease may be linked to impairment of nuclear RNA biogenesis and impact on DDR (Fig 6C). The application of RNA-based therapies to modulation of gene and subsequent protein expression is an attractive therapeutic strategy, that could be considered in the future for the treatment of ALS and other neurodegenerative diseases.

## Materials and methods

### Ethics statement

The study protocol was approved by the Ethical Committee of the IRCCS Mondino Foundation (Pavia, Italy). Subjects participating in the study signed an informed consent (Protocol n°375/04 –version 07/01/2004). The study conformed the standards of the Declaration of Helsinki.

## Human cell culture and transfection

HeLa cells (CCL-2, American Type Culture Collection [ATCC]) and SH-SY5Y (CRL-2266, American Type Culture Collection [ATCC]) were cultured in Dulbecco's modified Eagle medium (DMEM, Thermo Fisher Scientific) supplemented with 10% fetal bovine serum (FBS) (S1710-500, Biowest), 1% Penicillin/Streptomycin (L0022-100, Biowest), and 1% L-glutamine (11539876, Gibco). Murine hybridoma cell line ATCC HB-8730 were grown in DMEM supplemented with 10% fetal bovine serum, 4 mM L-glutamine, 4500 mg/L glucose, 1 mM sodium pyruvate, and 1500 mg/L sodium bicarbonate (5% $CO_2$). Lymphoblastoid cell lines (LCLs) were harvested with RPMI 1640 (42401018, Gibco) supplemented with 20% of FBS, 1% Penicillin/Streptomycin and 1% L-glutamine.

HeLa cells were transiently transfected using DharmaFect (Dharmacon) according to the manufacturer's instructions with siC (D-001810-01-05) and siTDP-43 (L-012394-00-0005) and collected after 72h, as described by Dominguez-Sanchez et al. [32]. HeLa cells and SH-SY5Y were stably transfected using Lipofectamine 2000 (Invitrogen, Carlsbad, CA), according to the manufacturer's instructions. HeLa cells were transfected with the following plasmids: pEGFP as control vector expressing GFP (Clontech), and pEGFP-M27, containing the GFP-fused RNaseH1 lacking the first 26 amino acids responsible for its mitochondrial localization cloned into pEGFP [64] that was used to overexpress RNase H1 [60]. All assays were performed 72 h after siRNA transfection or 24 h after plasmid transfection [41].

SH-SY5Y cells were stably transfected using Fugene HD Transfection Reagent (Promega) with C-terminally GFP-tagged TDP-43 WT vector (PS100010 pCMV6-AC-GFP), C-terminally GFP-tagged TDP-43 A382T mutation vector (CW303334 mutated ORF of RC210639 at nt position 1144, changed from G to A, to obtain A382T, inserted in PS100001) and a C-terminally GFP-tagged TDP-43 G294V mutation vector (CW303287 mutated ORF of RG210639 at nt position 881, changed from G to T, to obtain G294V, inserted in PS100010), purchased from OriGene. During the paper the SH-SY5Y are respectively named: SH-SY5Y expressing basal levels of TDP-43, SH-TDP+ overexpressing a GFP-tagged TDP-43 WT form, SH-TDP382 expressing the GFP-tagged p.A382T TDP-43 mutant form and SH-TDP294 expressing the GFP-tagged p.G294V TDP-43 mutant form.

## EBV inmortalization of cells from ALS patients

ALS diagnosis was made according to the revised El Escorial Criteria [65]. An healthy volunteer (male of 49 years old), free from any pharmacological treatment and pathology, was recruited at the Transfusion Centre of the IRCCS Policlinico S. Matteo Foundation in Pavia (Italy). Peripheral Blood Mononuclear Cells (PBMCs) from 2 ALS patients (one sporadic ALS patient not harboring mutations in the *SOD1*, *FUS/TLS*, *TARDBP*, *C9ORF72* and one ALS patient carrying a homozygous p.A382T TARDBP missense mutation) and 1 control were immortalized with EBV as previously described [46]. PBMCs were isolated from peripheral venous blood by Histopaque-1077 (Sigma-Aldrich) following the manufacturer's instructions. Briefly, $5 \times 10^6$ PBMC cells were re-suspended in RPMI 1640 medium (Sigma-Aldrich), supplemented with 20% fetal bovine serum (FBS; Sigma-Aldrich), 0.3 mg/l L-glutamine, 5% penicillin-streptomycin and cyclosporine A (Sigma-Aldrich). EBV-mix, prepared according to Caputo and collaborators [66], plus RPMI 1640 with cyclosporin A was added to the cells. Cells were incubated at 37˚C in a humidified atmosphere with 5% CO2 for 1 week. The medium was then changed and cells were left in incubation until clusters of growing cells appeared.

## Immunofluorescence microscopy

For S9.6 IF analysis in HeLa and SH-SY5Y, cells were fixed with cold methanol for 10 minutes at -20°C according the literature [38]. SH-SY5Y cells were treated with 40 U/ml RNaseIII (1 U/μl, Thermo Fisher Scientific) for 30 minutes at 37° using 1X RNase III Reaction Buffer. For γH2AX and FANCD2 IF analysis in HeLa and SH-SY5Y, cells were incubated with a fixation solution (PFA 4%, Triton-X 0,1%) for 10 minutes at room temperature (RT) as previously described [47]. For S9.6 and TDP-43 IF microscopy analysis in LCLs, cells were fixed with two methods: one using formaldehyde and cold acetone and the other using cold methanol.

The following antibodies were used: anti-nucleolin antibody (ab50279, Abcam), S9.6 mono-clonal antibody (ATCC HB-8730 hybridoma),anti-TDP-43 antibody (Clone: 6H6E12, Protein-tech), anti-γ-H2AX antibody (ab2893, Abcam), anti-FANCD2 antibody (sc-20022, SantaCruz), anti-H3S10P antibody (06–570, Sigma-Aldrich). Antibody signals were detected on a Leica DM6000 microscope equipped with a DFC390 camera (Leica). Data acquisition was performed with LAS AF (Leica). R-loops signal in the nucleoplasm of HeLa and SH-SY5Y cell lines was quantified using ImageJ program by measuring the S9.6 integrated density observed in the DAPI-stained nucleus, subtracting the nucleolar contribution, detected by nucleolin antibody. γH2AX and FANCD2 foci were determined in IF of HeLa and SH-SY5Y cell lines through the relative number of cells containing >10 foci in the nuclei in each condition. In each IF experiment we maintained around 100 counted cells for improving their comparability.

Nuclear area was obtained by measuring the DAPI area using FIJI (ImageJ) [67]

## DNA-RNA immunoprecipitation-qPCR

DNA-RNA immunoprecipitation (DRIP) was performed on HeLa and SH-SY5Y cells as already described in literature [58, 68]. The amount of R-loop levels was quantified as a function of input DNA, that for each sample was 10% of the entire amount. Primers used are listed in S1 Table.

## Flow cytometry analysis

LCLs from a control (LCL-CTL), a sporadic patient (LCL-SALS) and a A382T TDP-43 mutant patient (LCL-TDP382) were incubated with a vitality dye for 15 minutes (Zombie Violet Fixable Viability Kit, BioLegend). Then, LCLs were incubated for 20 minutes with anti-CD19 antibody for B lymphocytes recognition. Cells were fixed and permeabilized using a kit based on saponin permeabilization (Fixation/Permeabilization Solution Kit, BD) as described [69]. As negative control for R-loop presence, cells were treated with 60 U/ml of ribonuclease H (RNase H, 5.000 units/ml, NEB) using RNase H buffer at 37°C for 1 hour. As negative control for ssRNAs, cells were treated with 100 μg/mL ribonuclease A (RNase A, 10 mg/mL, Thermofisher) in 0.3M NaCl buffer at 37°C for 1 hour. At the end, cells were stained for one hour with conjugated anti-S9.6 antibody (PE/R-Phycoerythrin Conjugation Kit, Abcam) and analysed by flow cytometry (BD FACS Canto II). Logarithmic amplification was used for all channels and FACSDIVA was used for the analysis. Moreover, SH-SY5Y transfection efficiency was analysed through flow-cytometry, detecting cellular GFP signal.

## Co-immunoprecipitation studies

Chromatin fraction (Chr fraction) and whole lysate fraction (WL fraction) were extracted from LCL-CTL, LCL-SALS and LCL-TDP382 cells. Cells ($10^7$) were resuspended in buffer A (25 mM HEPES, 1.5 mM MgCl2, 10mM KCl, 0.5% NP-40, 1 mM DTT) incubated on ice for 10 minutes to isolate the nuclei and spin at 5000 rpm for 5 min at 4°C. The pellet was

resuspended in buffer B (50 μM HEPES pH 7.8, 140 mM NaCl2, 1mM EDTA, 1%Triton X-100, 0.1% Na-deoxycholate, 0.1% SDS) and sonicated for 5 cycles (30 ON/30 OFF) using a Bioruptor Sonicator Brochure—Diagenode. All buffer were supplemented with protease and phosphatase inhibitor. Magnetic beads (Dynabeads, Invitrogen) were conjugated with 10 μg S9.6 antibody and respective murine IgG antibody (sc-2025, Santa Cruz) for 2h at 4˚C in agitation, followed by incubation O/N with the samples, recovering 10% prior to incubation. After washes, a part of the samples was eluted using elution buffer (50mM Tris pH 8.0, 1mM EDTA, 1% SDS) at 65˚C for 10 minutes twice and resuspended in TE buffer for qPCR analysis. S9.6 immunoprecipitation was evaluated by qPCR in genomic sites enriched for R-loop presence in these cells (NOP58 and ING3) [70]. Input (10% fraction for each sample), S9.6 and IgG were loaded on a 10% SDS-PAGE and co-immunoprecipitation was evaluated by immunoblotting with rabbit anti-TDP-43 (1:1000 dilution), rabbit anti-H3 (1:1000 dilution), rabbit anti-GAPDH (1:10000 dilution). Anti-TDP-43 antibody (Clone: 6H6E12, Proteintech Europe), anti-H3 antibody (ab1791, Abcam), anti-GAPDH antibody (GTX100118, GeneTex), IgG antibody (sc-2025, Santa Cruz) were used.

## Quantitative PCR analysis

For real-time (RT)–qPCR analysis, cDNA was synthesized using QuantiTect Reverse Transcription Kit (Qiagen). mRNA expression values of the indicated genes were normalized with mRNA expression of the HPRT housekeeping gene. RT—qPCR was performed with iTaq Universal SYBR Green Supermix (Bio-Rad) and analyzed on 7500 FAST Real-Time PCR system (Applied Biosystems, Carlsbad, CA). Primers are listed in S1 Table.

## Viability assays

SH-SY5Y cell viability was assessed by Trypan Blue assay. Briefly, cell suspension was mixed with 0.4% Trypan Blue (Sigma-Aldrich) and counted in three independent experiments with the automated cell counter TC20 (Bio-Rad) to evaluate the percentage of live cells, which was about 75–80%.

## Western blot analysis

Total soluble protein samples were extracted in RIPA buffer [50 mM Tris-HCl, pH 8.0, 150 mM NaCl, 1% NP-40, 12 mM deoxycolic acid, supplemented with 1% protease inhibitor cocktail (PIC)]. Protein-containing supernatants were collected and stored at −80˚C. Total soluble protein lysates were loaded onto 8–16% SDS-PAGE gel (Bio-Rad). Samples were transferred to nitrocellulose membrane with Trans-Blot Turbo Transfer System (Bio-Rad). After 5% non-fat dry milk blocking, nitrocellulose membranes were incubated ON at 4˚C with the following antibodies: anti-TDP-43 antibody (Clone: 6H6E12, Proteintech), anti-GAPDH antibody (GTX100118, GeneTex). Densitometric analysis of the bands was performed using ImageJ software from three independent experiments.

SH-SY5Y subcellular fractionation was performed according to Pansarasa et al. 2018. Cells were first re-suspended in an ice-cold hypotonic lysis buffer [10 mM HEPES, pH 7.9, 10 mM KCl, 0.1 mM EDTA, 1 mM dithiothreitol (DTT), 1% PIC] and incubated on ice. 10% Nonidet NP-40 was added and samples were centrifuged at the maximum speed (16,000 *g*). Cytoplasmic proteins were collected and stored at −80˚C until use. The pellet was re-suspended in an ice-cold hypertonic nuclear extraction buffer (20 mM HEPES, pH 7.9, 0.4 M NaCl, 1 mM EDTA, 1 mM DTT, 1% PIC) and incubated on ice with agitation. The nuclear extracts were centrifuged at the maximum speed and the nuclear proteins were collected. GFP antibody (Santa Cruz Biotechnology(sc-390394, dilution 1:500), rabbit polyclonal anti-H1 (sc-67324 Santa Cruz Biotechnology;

dilution 1:250) and mouse monoclonal anti-lactate dehydrogenase (LDH; sc-133123 Santa Cruz Biotechnology; dilution 1:1000) were used as nuclear and cytoplasm loading controls.

## Genome-wide data collection

TDP43 ChIP-seq data was obtained from ENCODE project (www.encodeproject.org) ENCSR033VAZ entry, while DRIPc-seq, DRIP-seq and RNA-seq was gathered from Gene Expression Omnibus under accession number GSE127979 [42]

## Genome-wide data downstream analysis

TDP43 ChIP-seq optimal IDR thresholded peaks from 2 biological replicates were retrieved from ENCODE (ENCFF909RMQ). DRIP-seq and DRIPc-seq sequencing reads (GSE127979) were mapped to the human hg38 canonical reference genome using Bowtie2 [71] (Langmead & Salzberg, 2012), while HISAT2 [72] (Kim et al., 2019) was used in the case of RNA-seq. Coverage profiles and metaplot images were obtained using deeptools 2 [73] (Ramírez et al., 2016). MACS2 package [74] (Zhang et al., 2008) was used for peak calling (FDR<0.01; broad region detection with 0.1 cutoff) on DRIPc-seq data. Only regions presenting peaks in both replicates were considered and merged when closer than 5kb using BEDtools [75] (Quinlan & Hall, 2010). In RNA-seq analysis, expressed genes where considered those ones presenting an RPKM average >0.001 from both replicates. ChIPseeker package [76] (Yu etal., 2015) with Ensembl genome annotation release 94 2018 88 [77] (Zerbino et al., 2018) was used for peak annotation. Venn diagrams were obtained using the BioVenn application [78] (Hulsen, T. et al. 2008).

## Statistical analysis

Statistical analysis was performed by Student t-test and by One-Way Analysis of Variance (ANOVA test) followed by post hoc comparison as a post-test (GraphPad Prism version 5, San Diego, CA, USA), unless otherwise specified. Values were considered statistically significant when p values were < 0.05.

## Supporting information

**S1 Fig. R-loop accumulation in siTDP-43 HeLa. Related to Fig 1. A) Relative** TDP43 expression as measured by cDNA-qPCR in control (siC) and TDP-43-depleted (siTDP43) cells. Unpaired t test, two-tailed was done. **B)** Percentage of cells efficiently transfected (GFP+) overexpressing RNH1. **C)** Comparative table on S9.6 FC (fold change) between results presented in this study for TDP-43 and others reported for factors involved in R loop homeostasis in HeLa cells. Table shows the relative increase in S9.6 reactivity (S9.6 FC), the cell process in which the factor is involved and the reference. **D)** DRIP-qPCR at the SNRPN (negative control) gene in siC and siTDP-43-treated HeLa cells gDNA untreated (-) and treated (+) with RNH. Unpaired t test, one-tailed was done. For all cases: Blank, not significant; *, P < 0,05; **, P < 0,01; *** P < 0,001; ****, P < 0,0001.
(TIF)

**S2 Fig. Analysis of TDP-43 protein expression in SHSY-TDP382 and SHSY-TDP294 cells. Related to Fig 4. A)** WB against GFP, TDP-43 and GAPDH in SH-SY5Y, SH-TDP+, SH-TDP382 and SH-TDP294. Relative quantifications of TDP43 overexpression rate, measured as GFP-TDP-43/GAPDH, are indicated in the histogram (n = 3). Unpaired t test, two-tailed was done. **B)** Western blot analysis of nuclear and cytoplasmic fractions of endogeneous TDP-43 and overexpressed TDP-43 forms fused to GFP. Mean+SEM (n = 3) are indicated in the histogram. *, P < 0,05; **, P < 0,01 (Unpaired t test, two-tailed). **C)** Flow cytometry

analysis of TDP-43 overexpression. GFP intensity (GFP-TDP43) for each cell line is shown. Percentage of cell over the threshold are indicated. **D)** TDP-43 overexpression efficiency as measured by IF. Representative images are showed in the left part, while scatter plot in the right presents TDP43 overexpression rates, as measured by total GFP intensity (GFP-TDP43) in cell population. Median values are indicated. Scale bar: 25μm. Mann-Whitney U test, two-tailed was done. **E)** IF using TDP-43 and GFP antibodies on SH-SY5Y, SH-TDP+ and SH-TDP382. TDP-43 nuclear abundances for each cell line, measured as total TDP43 intensity, are indicated in the scatter plot. Median values are indicated. Scale bar: 25μm. Mann-Whitney U test, two-tailed was done. **F)** Nuclear area in SH-SY5Y, SH-TDP+ and SH-TDP382. Mean +SEM (n = 3) are indicated in the histogram. Other details as in Fig 4 and S1 Fig. (TIF)

**S3 Fig. R-loops accumulate in SHSY-TDP382 and SHSY-TDP294 cells. Related to Fig 4. A)** Representative images of S9.6 and nucleolin IFs in SH-SY5Y, SH-TDP+, SH-TDP382 and SH-TDP294. Scale bar: 25μm. **B)** Percentage of cells efficiently transfected (GFP+) overexpressing RNH1. Unpaired t test, two-tailed was done. **C)** DRIP-qPCR at the SNRPN (negative control) gene in SH-SY5Y, SH-TDP+ and SH-TDP382 cells gDNA untreated (-) and treated (+) with RNH. Other details as in Fig 4 and S1 Fig. (TIF)

**S4 Fig. R-loops accumulate in p.A382T TDP-43 mutated LCLs. Related to Fig 6. A)** IF of LCL-CTL, LCL-TDP382, LCL-SALS using an anti-TDP-43 antibody and an anti-S9.6 antibody after paraformaldehyde fixation. The scatter plots show the increase of S9.6 signal intensity and the decrease in TDP43 nuclear content in SH-TDP382. Median values are indicated. Scale bar: 10μm. Mann-Whitney U test, two-tailed was done. Nuclear area in SH-SY5Y, SH-TDP+ and SH-TDP382 when fixed in paraformaldehyde **(B)** and methanol **(C)** are indicated. Median+SEM are indicated in the histogram. Unpaired t test, two-tailed was done. Other details as in Fig 6 and S1 Fig. (TIF)

**S5 Fig. Cytoplasmic mislocalization of TDP-43 in p.A382T TDP-43 SH-SY5Y. Related to Fig 6.** TDP-43 and TDP-35 strongly interact with S9.6 antibody in TDP-43 mut LCLs WL fraction. **A)** and **B)** CoIP between S9.6 and TDP-43 in chromatin of LCL-CTL, LCL-TDP382, LCL-SALS. Input, S9.6 IP and IgG IP of chromatin fraction were loaded on a 10% SDS-PAGE and then immunoblotted with TDP-43, H3 and GAPDH as nuclear and cytosolic loading control. S9.6 binding was tested by qPCR. Quantification of TDP43 relative amounts in chromatin and whole lysate co-IPs are indicated in the histograms. Mean+ SEM are indicated. Unpaired t test, two-tailed was done. Other details as in Fig 6 and S1 Fig. (TIF)

**S1 Table. Oligonucleotides used in this study.**
(DOCX)

**S1 Data. Source Data: Spreadsheet of source data shown in this study.**
(XLSX)

## Author Contributions

**Conceptualization:** Cristina Cereda, Andrés Aguilera.

**Formal analysis:** Marta Giannini, Aleix Bayona-Feliu, Daisy Sproviero, Cristina Cereda, Andrés Aguilera.

**Funding acquisition:** Cristina Cereda, Andrés Aguilera.

**Investigation:** Marta Giannini, Daisy Sproviero, Sonia I. Barroso.

**Methodology:** Marta Giannini, Aleix Bayona-Feliu, Daisy Sproviero.

**Project administration:** Cristina Cereda, Andrés Aguilera.

**Resources:** Aleix Bayona-Feliu, Andrés Aguilera.

**Supervision:** Aleix Bayona-Feliu, Daisy Sproviero, Cristina Cereda, Andrés Aguilera.

**Visualization:** Aleix Bayona-Feliu, Daisy Sproviero.

**Writing – original draft:** Marta Giannini, Cristina Cereda, Andrés Aguilera.

**Writing – review & editing:** Marta Giannini, Aleix Bayona-Feliu, Daisy Sproviero, Cristina Cereda, Andrés Aguilera.

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
