## [Decision Letter · Decision Letter 0]

17 Jun 2020

Dear Dr Aguilera,

Thank you very much for submitting your Research Article entitled 'TDP-43 deficiency links Amyotrophic Lateral Sclerosis with R-loop homeostasis and R loop-mediated DNA damage' to PLOS Genetics. Your manuscript was fully evaluated at the editorial level and by independent peer reviewers. The reviewers appreciated the attention to an important problem, but raised some substantial concerns about the current manuscript. While Reviewer 2 was generally positive, Reviewers 1 and 3 found that the presented data are not sufficient to support the manuscript's conclusions.  Both reviewers indicated and Reviewer 3 specifically suggested using additional experimental approaches to validate your conclusions.  Also, please note a request of Reviewer 1 to include source data underlying graphical display items, which is currently a PLOS Minimal Data Set Definition requirement.  (see below the link and clarification of data availability policy).

Based on the reviews, we will not be able to accept this version of the manuscript, but we would be willing to review again a much-revised version. We cannot, of course, promise publication at that time.

If you decide to revise the manuscript for further consideration at PLOS Genetics, please aim to resubmit within the next 60 days, unless it will take extra time to address the concerns of the reviewers, in which case we would appreciate an expected resubmission date by email to plosgenetics@plos.org.

[LINK]

We are sorry that we cannot be more positive about your manuscript at this stage. Please do not hesitate to contact us if you have any concerns or questions.

Yours sincerely,

Dmitry A. Gordenin

Associate Editor

PLOS Genetics

Gregory Barsh

Editor-in-Chief

PLOS Genetics

Reviewer's Responses to Questions

**Comments to the Authors:**

Reviewer #1: In the manuscript “TDP-43 deficiency links Amyotrophic Lateral Sclerosis with R-loop homeostasis and R loop-mediated damage,” Giannini and co-authors explore the role of TDP-43 in the maintenance of genome integrity and try to dissect the mechanism of the neurodegenerative pathogenesis caused by the mutated version of the protein. They test the hypothesis that TDP-43 prevents the accumulation of DSB-inducing, deleterious R-loops and claim that an R-loop-dependent increase in DSB foci and Fanconi Anemia repair centers is caused by mis-localization of the mutated protein in transfected neuronal cells and in cell lines from ALS patients. Without a doubt, the findings reported in the manuscript are significant and worth further investigation; however, the conclusions drawn from the data are not sufficiently substantiated.

Major concerns.

1. The depletion of TDP-43 in HeLa cells by siRNA leads to a minor increase in S9.6 signal intensity (Figure 1A). Even though the increase is statistically significant, the biological relevance of such an increase is unclear without comparison to the changes in intensity after depletion of the known protein involved in R-loop processing. This concern is exacerbated by the absence of an increase in DRIPqPCR product for the APOE gene in siTDP-43 treated cells (Figure 1B). Taking into account this marginal effect on R-loop formation on the one hand and the multiple functions of TDP-43 in mRNA processing and its interaction with DNA on the other, the increase in gammaH2AX and FANCD2 foci (Figure 2) could reflect consequences of TDP-43 depletion that are unrelated to R-loop processing. The decrease in foci in cells expressing RNaseH could also be an indirect consequence of the protein depletion.

2. The statement from the abstract, “we prove that mislocalization of mutated TDP-43 (A382T) in transfected neuronal SH-SY5Y and lymphoblastoid cell lines (LCLs) from an ALS patient cause R-loop accumulation, and R loop-dependent increased DSBs and Fanconi Anemia repair centers” is misleading. Mis-localization of both proteins - mutant and wild type - leads to an increase in S9.6 intensity (Figure 3A), and the difference in S9.6 signal intensity between the cells with mutant and wild type proteins is not significant. It is well documented that overexpression of proteins can cause their mis-localization and misfunction. So, “significantly decreased” nuclear localization of the mutant protein in SHSY-TDP382 cells (decrease of less than 15%) compared to wild type SHSY-TDP+ cells (Supplementary Figure 1) can result from (slightly) different expression levels of each of the proteins in the corresponding cell lines. In other words, the somewhat increased deleterious consequences of expression of the mutant allele could be caused by a higher level of overexpression of the mutant protein, compared to the wild type allele, rather than the presence of a missense mutation.

Unless the levels of TDP+ and TDP382 proteins are measured and found similar, the “mutation” effects could be attributed to the differential expression levels of TDP-43 alleles in the cells.

3. The same critique applies to the experiments with the lymphoblastoid cells from the patients.

4. The data in Figure 6 are neither quantified nor normalized. The quantity of the loading control proteins varies from lane to lane. Also, it is unclear how many times the experiment has been performed, so the discussion of the results of this experiment is mere speculation.

5. In Figure 3A, the increase in S9.6 signal in the cells with overexpression of TDP+ cannot be attributed to R-loop formation because overexpression of RNaseH does not decrease the signal. Nevertheless, the authors make the following conclusion: “Therefore, we conclude that mislocalisation of TDP-43 also causes R-loop accumulation “(page 8). This statement contradicts the data. It is unclear whether the levels of overexpression of RNaseH were monitored during the experiments in different cell lines. One explanation of these data could be that the expression of RNaseH in the specific cell line was different (lower) from the others.

6. Throughout the study, the only assay utilized for providing evidence for DNA damage or activation of DNA damage response is the detection of DNA repair foci. Additional, independent tests (such as measurement of homologous recombination or non-homologous end-joining in reporter systems, evaluation of checkpoint activation, assessment of sensitivity to DNA damaging agents) are necessary to address physiological consequences of TDP43 malfunctioning.

Minor concerns.

1. There is no description of how overexpression of RNaseH is achieved in Figures 1-5. Reference #33, used for an explanation (Ginno et al., 2013), does not provide relevant information.

2. Since in Figures 3A and Supplemental Figure 1, the data presented as “signal per nucleus” and the differences in these signals are minor, it seems to be important to make sure that the area (size) of the nuclei are similar among cells transfected with different constructs.

3. Flow cytometry plots in Figures 5B and C do not include the plot of S9.6 signal for the LCL-SALS cells (cells from sporadic ALS patient), making the data representation skewed.

Reviewer #2: In this manuscript, Gianini et al. have investigated in detail the role played by TDP-43 in the generation of R-loops and RNA:DNA hybrids in neuronal and non-neuronal cell lines. In general, the manuscript contains interesting data for the field and several novel observations that contribute to a better understanding of the protein in disease. A few required additions and clarifications are as follows:

-The title “TDP-43 deficiency links Amyotrophic Lateral Sclerosis with R-loop homeostasis and R loop-mediated DNA damage” does not seem appropriate since the manuscript principally focuses on a disease-associated mutation, A382T. Therefore, it might be better to change it in “TDP-43 mutation links Amyotrophic Lateral Sclerosis with R-loop homeostasis and R loop-mediated DNA damage”.

- The order followed in the abstract is not the one respected in the results (ie. in the abstract the authors first write about mislocalization of mutated TDP-43 (A382T) in transfected neuronal SH-SY5Y and lymphoblastoid cell lines (LCLs) from an ALS patient cause R-loop accumulation and then a non-neuronal model of HeLa cells depleted of TDP-43. However, in the results they present first the depletion of TDP-43 in Hela cells and then the transfection of mutated TDP-43 in SHSY-5Y. Please be consistent in the order of the text.

- In the work the authors used silenced TDP-43 in Hela cells but transfected mutated TDP-43 in SHSY-5Y. Please explain better the choice of the two cell lines and how they are connected.

- In the LCLs mutated in TDP-43 the authors observe an increase of R-loops compared to CTL and SALS LCLs. RNAse H could revert partially the S9.6 increase signal. How can this partial decrease be explained?

- Experiments were run in SHSY5Y and in LCL carrying only one mutation A382T. It might be interesting to carry out at least part of the reported experiments in SHSY-5Y cells overexpressing TDP-43 with another mutation. These data would confirm whether the readouts observed with this mutation are specific or not.

Reviewer #3: The paper by Gianini et al focuses on understanding how deficiency of TDP-43 observed in ALS is connected with R-loop homeostasis and R-loop mediated DNA damage. Previous work from Livinsgton lab has demonstrated that TDP43 is involved in prevention/repair of transcription-associated DNA damage through R-loop-related mechanisms in neuronal cells (Hill et al PNAS 2016), therefore the novelty of this paper is partially compromised by this publication. One interesting aspect which the authors were trying to pursue is to uncover if TDP-43-disease-associated mutations which result in re-localisation of usually nuclear TDP43 to the cytoplasm has any effect on R-loop-mediated DNA damage. This is certainly an interesting question worth addressing. The authors pursued these questions in HeLa cells (where TDP43 is depleted, Fig 1-2), neuroblastoma cells (where TDP-43 wt or mutant is over-expressed, Fig 3-4) and lyphoblastoid cells (with patient mutations in endogenous TDP43, Fig 5-6). Using these model systems, the authors have shown that TDP43 deficiency results in accumulation of R-loops (by IF and DRIP-qPCR), DNA damage (γH2AX) and Fanconi Anemia repair centers. Unfortunately, the authors have not managed to take these initial findings, some of which were already reported in Hill et al PNAS 2016. (i.e. R-loop and DNA damage increase), into subsequent mechanistic investigations. For the reasons of the quality and amount of data presented in this paper, at this current stage the paper is not suitable for publication in PLoS Genetics.

Major comments:

1. The amount of data presented in this paper is minimal and not of a good quality which overall does not give me the confidence of the general paper conclusions. In particular the conclusions drawn from S9.6 IF analysis are rather ‘pushed’ and not visible from the images presented. For example, Fig 1A does not have a clear nuclear signal in any of the images shown (only nucleolar signal is visible). The differences between quantified signals are very small and represent only a background signal, in my opinion). The RNase H over-expression does not result in any visible change of the signal on the images. The positive controls (known factor which up-regulates R-loop level) are missing. The authors have not quantified nucleolar signal (are there any changes there?), though they have chosen to study gene encoding ribosomal protein in their DRIP analysis (see below).

2. It is surprising to me that the only gene which shows statistically significant accumulation of R-loops is RPL13A, which is a ribosomal protein. I would like to see more comprehensive analysis of R-loop increase by DRIP-qPCR. Other multiple genes of different categories should be tested in different model systems.

3. Experiments in Fig 3-4 are carried out with neuroblastoma cells, where TDP-43 Wt or mutant (TDP382) are expressed on the endogenous background. The authors need to characterise this system a bit better, since the imaging in Fig S1 is not sufficient. They need to present Westen blot showing the level of expression of endogenous and over-expressed proteins in different cellular fractions (i.e. nucleus, cytoplasm) in all three cell lines. What is the level of endogenous protein contribution to all results observed? The authors have not really explained why do we observe a significant decrease of nuclear signal for TDP43 WT cells compared to control cells? These data are important for further interpretation of IF and DRIP results. All these experiments should have been performed on the background of depleted endogenous TDP-43 protein to have a clear interpretation.

4. The authors have carried out IF in Lymphoblast cells from patients with TDP43 and ALS mutations. Similar to my comments for Figures 1A and 3A, the IF data are not convincing and show extremely small changes between cell lines studied. These cells derived from different people and are subject to patient –to patient genetic variation. Since the data are showing rather small changes, the authors should use multiple cell lines representing each category. Possibly using alternative approaches such as slot blot may provide more quantitative data than IF.

5. Figure 6 contains results of S9.6 IP experiments in lymphoblast cells. The authors propose that they observe less TDP43 interaction with R-loops when TDP43 is mutated. This is not visible from the protein gel presented, and as far as I am concern represents a wishful thinking of the authors and not the result of a given experiment. There is absolutely minimal pull-down observed with TDP-43 in the nucleus, and mainly there is a pull down of TDP-35 which should be really cytoplasmic, and not in the chromatin fraction? Clearly these TDP-35 fragments are also IPed in the WT conditions – why is this happening? This experiment lacks RNase H control to show that interaction between TDP43 and R-loops is specific. Comments regarding stronger pull-down of TDP35 in TDP43 mutant in cytoplasm are not really substantiated since it is clear that this sample is over-loaded on the gel as can be seen from H3 and GAPDH westerns.

Minor comments:

1. On page 9 first paragraph, the authors say that ‘TDP-43 mutation causes an increase in DNA breaks derived from R-loop accumulation that promotes transcription-replication collisions..’ . I think that this is an over-statement since it has not really been tested if accumulation of R-loops following TDP-43 depletion results in transcription-replication collisions.

2. The authors need to explain why SNRPN gene does not have DRIP signal – it is not expressed?

3. All the data in this paper can be grouped in 2-3 figures max (one figure for each cell model used).

4. The paper would benefit from additional English editing (i.e. - ‘significant’ on pages 6 and 11 should be spelled correctly..etc)

5. Concluding model demonstrating TDP43 role in disease and R-loop biology would be helpful.

**Have all data underlying the figures and results presented in the manuscript been provided?**

Reviewer #1: No: The numerical data underlying graphs are not provided in spreadsheet form.

Reviewer #2: Yes

Reviewer #3: Yes

PLOS authors have the option to publish the peer review history of their article (what does this mean?). If published, this will include your full peer review and any attached files.

Reviewer #1: No

Reviewer #2: No

Reviewer #3: No

---

## [Decision Letter · Decision Letter 1]

2 Nov 2020

Dear Dr Aguilera,

Thank you very much for submitting your Research Article entitled 'TDP-43 mutations link Amyotrophic Lateral Sclerosis with R-loop homeostasis and R loop-mediated DNA damage' to PLOS Genetics. Your manuscript was fully evaluated at the editorial level and by independent peer reviewers. The reviewers appreciated the attention to an important topic but identified some aspects of the manuscript that should be improved.

We therefore ask you to modify the manuscript according to the review recommendations before we can consider your manuscript for acceptance. Your revisions should address the specific points made by each reviewer.

1) Provide a detailed list of your responses to the review comments and a description of the changes you have made in the manuscript.  Please also note that one of reviewers stated that the revision should include all source data, i.e., numerical data underlying graphs or summary statistics are included with the submission.  This request goes along with PLOS data availability policy.  Should your manuscript is accepted for publication, the most helpful to future readers would be if the specific sections of supplemental tables or separate supplementary tables containing source data for the display item would be specified in the corresponding figure legend or table footnotes.

[LINK]

Yours sincerely,

Dmitry A. Gordenin, Ph.D.

Associate Editor

PLOS Genetics

Gregory Barsh

Editor-in-Chief

PLOS Genetics

Reviewer's Responses to Questions

**Comments to the Authors:**

Reviewer #1: The revised manuscript by Gianini et al., “TDP-43 deficiency links Amyotrophic Lateral Sclerosis with R-loop homeostasis and R loop-mediated damage,” is much improved compared to the original version.

Most of the issues that caused concerns in the paper’s previous version regarding data interpretation and appropriate controls are addressed and resolved in the updated manuscript.

The added comparison of the consequences of TDP-43 depletion on R-loop formation (reported as changes in S9.6 fluorescence intensity) to the measurements already published in the studies of the contribution of proteins directly involved in R-loop processing (S1C Fig), as well as DRIP-seq of additional R-loop - prone targets, allows appreciation of the biological relevance of relatively minor changes in fluorescence levels.

The data from the new experiments involving treatment of the cells with RNAse III for the removal of dsRNA (Figure 4A) eliminate the ambiguity of the results reported in the first version of the manuscript and validate the statement, “we prove that mislocalization of mutated TDP-43 (A382T) in transfected neuronal SH-SY5Y and lymphoblastoid cell lines (LCLs) from an ALS patient cause R-loop accumulation…”

Measurements of the expression levels of the wild type and mutant constructs (reported in S2A and C Figures) strengthen the conclusion that indeed mislocalization, rather than an increased level of the mutant protein, causes accumulation of R-loops.

Several points still require the authors’ attention.

1. Even though the new genome-wide analysis of TDP-43 localization (Figure 3) does not provide direct evidence of the contribution of TDP-43 to R-loop processing but instead suggests that TDP-43 binds to highly expressed loci in DNA, these data offer new insights into TDP-43 functions. Nevertheless, the statement “…TDP-43 RBP controls transcription at expressed genes that are prone to accumulate R loops...” is not substantiated by the data and should be toned down.

2. Since statistically significant “reduction in nuclear fraction” of endogenous TDP-43 was NOT observed either following expression of SH-TDP+, or SH-TDP382, it is formally incorrect to attribute an absence of a reduction in SH-TDP294 to any phenotype (S2BFig). (“In the case of SH-TDP294, a reduction in the nuclear fraction was not observed in the western blot, therefore implying that protein dysfunction can be causative of the phenotype.”, p. 8).

3. The model in Figure 6C must be adjusted to accommodate the data showing that a decrease (compared to control) in nuclear localization of overexpressed wild type TDP-43 (2SE Figure, SH-TDP+ vs. SH-SY5Y) does not cause an increase in S9.6 intensity (Figure 4A).

4. The spreadsheets with the raw data reflected in graphs and figures are not provided.

Reviewer #2: The authors have provided several new data and have addressed in detail most of the queries raised by reviewers.

Reviewer #3: The authors have made an effort to improve this manuscript including additional genes for DRIP-qPCR analysis (as requested in my comment 2) and some modest visual improvement for S9.6 IF.

Related to previous comment 1:

To address my concern related to small changes in IF signal in the nucleus associated with Pol II genes, the authors introduced genomic data from publicly available Encode and previously published data, showing a correlation between TDP-43 binding and R-loops in expressed genes under physiological conditions in K562 cells (fig 3). Based on these datasets TDP-43 is mainly enriched at the 5’end /TSS of the expressed genes, while R-loops can be found in multiple locations (as shown by the DRIP-qPCR examples in the paper, where TDP-43-sensitive R-loops are enriched across various genomic locations). However, these new genomic data in Fig 3 does not clarify what happens with this correlation when TDP-43 is depleted? Do R-loops accumulate only at the locations where TDP-43 is bound which would support the direct effects of TDP-43 on the regulation of these R-loops? The authors have not demonstrated if they see a specific enrichment of TDP-43 binding at R-loops of their tested genes by DRIP-qPCR supporting the specificity and direct effects of TDP-43 on these R-loops.

Related to previous comment 4:

Unfortunately the authors have failed to introduce additional patient cell lines to verify the effects of R-loop accumulation in patients with different genetic backgrounds. Therefore, in my opinion, providing a direct evidence that TDP-43 binds at the regions of R-loops affected following its depletion (as discussed above) is crucial.

**Have all data underlying the figures and results presented in the manuscript been provided?**

Reviewer #1: **No: **The data underlying the figures are not provided

Reviewer #2: Yes

Reviewer #3: Yes

PLOS authors have the option to publish the peer review history of their article (what does this mean?). If published, this will include your full peer review and any attached files.

Reviewer #1: No

Reviewer #2: No

Reviewer #3: No

---

## [Editor Report · Decision Letter 2]

8 Nov 2020

Dear Dr Aguilera,

We are pleased to inform you that your manuscript entitled "TDP-43 mutations link Amyotrophic Lateral Sclerosis with R-loop homeostasis and R loop-mediated DNA damage" has been editorially accepted for publication in PLOS Genetics. Congratulations!

Yours sincerely,

Dmitry A. Gordenin, Ph.D.

Associate Editor

PLOS Genetics

Gregory Barsh

Editor-in-Chief

PLOS Genetics

Comments from the reviewers (if applicable):

**Data Deposition**

http://datadryad.org/submit?journalID=pgenetics&manu=PGENETICS-D-20-00749R2

**Press Queries**

---

## [Editor Report · Acceptance letter]

28 Nov 2020

PGENETICS-D-20-00749R2 

TDP-43 mutations link Amyotrophic Lateral Sclerosis with R-loop homeostasis and R loop-mediated DNA damage 

Dear Dr Aguilera, 

We are pleased to inform you that your manuscript entitled "TDP-43 mutations link Amyotrophic Lateral Sclerosis with R-loop homeostasis and R loop-mediated DNA damage" has been formally accepted for publication in PLOS Genetics! Your manuscript is now with our production department and you will be notified of the publication date in due course.

With kind regards,

Nicola Davies

PLOS Genetics

On behalf of:
